# Technical note: Influence of different averaging metrics and temporal resolutions on aerosol pH calculated by thermodynamic modeling

Haoqi Wang[1,2,★], Xiao Tian[1,2,★], Wanting Zhao[1,2], Jiacheng Li[1,2], Haoyu Yu[1], Yinchang Feng[1,2], Shaojie Song[1,2]

[1]State Environmental Protection Key Laboratory of Urban Ambient Air Particulate Matter Pollution Prevention and Control, College of Environmental Science and Engineering, Nankai University, Tianjin 300350, China
[2]CMA-NKU Cooperative Laboratory for Atmospheric Environment-Health Research, Tianjin 300350, China
★These authors contributed equally to this work.

*Correspondence to*: Shaojie Song (songs@nankai.edu.cn)

**Abstract.** Aerosol pH is commonly used to characterize the acidity of aqueous aerosols and is of significant scientific interest due to its close relationship with atmospheric processes. Estimation of ambient aerosol pH usually relies on the thermodynamic modeling approach. In the existing chemical transport model and field observation studies, the temporal resolution of the input chemical and meteorological data into thermodynamic models varied substantially ranging from less than an hour to a year because of the inconsistency in the resolution of the original data and the aggregation of time-series data in some studies. Furthermore, the average value of aerosol pH has been represented by diverse metrics of central tendency in existing studies. This study attempts to evaluate the potential discrepancies in the calculated average aerosol pH arose from differences in both averaging metrics and temporal resolutions based on the ISORROPIA-II thermodynamic model and the example datasets prepared by the GEOS-Chem chemical transport model simulation. Overall, we find that the variation in the temporal resolutions of input data may lead to a change of up to more than two units in the average pH, and that the averaging metrics calculated based on the pH value of individual samples may be about two units higher than the averaging metrics calculated based on the activity of hydrogen ions. Accordingly, we recommend that the chosen averaging metrics and temporal resolutions should be stated clearly in future studies to ensure comparability of the average aerosol pH between models and/or observations.

## 1 Introduction

Aerosol acidity, typically characterized by pH, stands as a crucial property of aqueous aerosols, influencing various physical and chemical processes (Tilgner et al., 2021). Aerosol pH can influence aerosol mass by regulating the gas−particle partitioning of semi-volatile acids and bases such as $HCl-Cl^-$, $HNO_3-NO_3^-$, and $NH_3-NH_4^+$ (Zhang et al., 2021a; Nah et al., 2018; Ding et al., 2019), as well as by moderating the production of secondary components through multiphase reactions (Pye et al., 2013; Cheng et al., 2016). Aerosol pH can also affect the solubilization of trace metals such as copper and iron, and therefore has

implications for human health risks and nutrient cycling in ecosystems (Meskhidze et al., 2003; Lippmann, 2014; Vasilakos et al., 2018; Wu et al., 2023).

The definition of pH is the negative log (base 10) of hydrogen ion (H⁺) activity on a molality basis according to the International Union of Pure and Applied Chemistry (IUPAC, https://goldbook.iupac.org/terms/view/P04524, last access: 15 February 2024), as shown in Eq. (1)

$$\text{pH} = -\log_{10}(a_{H^+}) = -\log_{10}(\gamma_{H^+} \frac{m_{H^+}}{m^{\ominus}}) = -\log_{10}(\frac{m_{H^+}}{m^{\ominus}}) - \log_{10}(\gamma_{H^+}) \tag{1}$$


where $a_{H^+}$ is the activity of H⁺ (dimensionless), $m^{\ominus}$ is the standard molality (1 mol kg⁻¹ water), and $m_{H^+}$ and $\gamma_{H^+}$ indicate the molality (mol kg⁻¹ water) and the activity coefficient (dimensionless) of H⁺, respectively. pH is also frequently defined based on the standard state of 1 mol H⁺ dm⁻¹ solution (i.e., molarity based) or the standard state of a hypothetical pure H⁺ solution (i.e., mole fraction based). Jia et al. (2018) comprehensively compared aerosol pH quantified at the three different

scales (i.e., molarity, molality, and mole fraction). They found that the difference between the mole-fraction-based and the molality-based pH values is a constant equal to 1.74. A minor difference (< 0.25 unit) exists between the molarity-based pH and the molality-based pH values owing to the effects of temperature, pressure, and the composition and density of aerosols.

The pH values of ambient aerosols are generally estimated depending upon the thermodynamic modeling approach since a commonly accepted direct measurement method is still lacking despite some recent significant advances (Weber et al., 2016;

Li and Kuwata, 2023; Li et al., 2023; Cui et al., 2021; Ault, 2020) . Thermodynamic models can predict the gas−particle, solid−liquid, liquid−liquid equilibria, liquid-phase activity coefficients, mass transfer of semi-volatile species, aerosol liquid water content (AWC, μg m⁻³ air), and pH (Pye et al., 2020). The input data for thermodynamic modeling should include the total (gas plus particle) chemical compositions (e.g., HCl + Cl⁻, HNO₃ + NO₃⁻, and NH₃ + NH₄⁺) and meteorological variables (relative humidity (RH) and temperature (T)). The E-AIM, AIOMFAC-GLE, MOSAIC, and ISORROPIA-II thermodynamic

models are the common box models used to calculate aerosol pH, which differ by their treated chemical species, computational complexity and rigor, and solution methods for activity coefficients. MOSAIC and ISORROPIA-II are computationally efficient for application in three-dimensional chemical transport models such as WRF-Chem, WRF-CMAQ, and GEOS-Chem (Pye et al., 2020).

Many studies have calculated aerosol pH using the thermodynamic modeling approach (see a brief summary in Table S1).

The chemical and meteorological input data were obtained from either three-dimensional chemical transport model simulations or field observations. In these studies, the temporal resolution of the input data varied substantially ranging from less than an hour to a year. The original time resolutions of field observations may be from tens of minutes to one week. Chemical transport model simulations usually have time resolutions of less than an hour. Some studies may aggregate the time series of the original chemical and meteorological data to a lower resolution (monthly, seasonally, or yearly) before running thermodynamic models,

while the others use the original dataset as model input.

The average of the aerosol pH dataset obtained from thermodynamic modeling has been represented by diverse measures of central tendency in existing studies (Table S1). A common metric was the arithmetic mean, denoted by $\overline{\text{pH}}$ and calculated with Eq. (2). The symbols $n$ and $i$ are the number of samples in the dataset and the $i^{\text{th}}$ sample, respectively. Another two metrics, $\overline{\text{pH}}^*$ (pH based on the arithmetic mean of $a_{\text{H}^+}$) and $\overline{\text{pH}}_{\text{w}}^*$ (pH based on the AWC-weighted mean of $a_{\text{H}+}$), have also been employed in previous studies to represent the center of the aerosol pH dataset, as described in Eqs. (3) and (4), respectively. Similar to $\overline{\text{pH}}_{\text{w}}^*$, the pH based on the volume-weighted mean of [$\text{H}^+$] has been usually considered when averaging cloud/fog water pH and precipitation pH (MÖller and Zierath, 1986; Sun et al., 2010; Straub et al., 2012; Shah et al., 2020).

$$\overline{\text{pH}} = \frac{1}{n}\sum_{i=1}^{n}\text{pH}_i, \tag{2}$$

$$\overline{\text{pH}}^* = -\log_{10}\left(\frac{1}{n}\sum_{i=1}^{n}\left(a_{\text{H}^+}\right)_i\right), \tag{3}$$

$$\overline{\text{pH}}_{\text{w}}^* = -\log_{10}\left(\frac{\sum_{i=1}^{n}\left\{\left(a_{\text{H}^+}\right)_i(\text{AWC})_i\right\}}{\sum_{i=1}^{n}(\text{AWC})_i}\right), \tag{4}$$

Since pH and $a_{\text{H}^+}$ are both non-conservative quantities upon mixing of individual samples, different averaging metrics and different temporal resolutions may lead to disparate values, posing potential challenges when comparing the reported average pH across studies. However, such discrepancies have not been addressed with sufficient care. The objective of this technical note is thus to quantitatively assess the averaged aerosol pH values using different metrics and different temporal resolutions. The rest of this article is structured as follows. In the Methods section (Sect. 2), we describe the preparation of the evaluation datasets in Sect. 2.1, which include the relevant chemical and meteorological variables and are obtained from the GEOS-Chem chemical transport model simulations. Statistical methods and analytical tools are provided in Sect. 2.2. In the Results and Discussion section (Sect. 3), we first present the probability distributions of aerosol pH and AWC from the evaluation dataset and estimate the differences among averaging metrics (Sect. 3.1). Sect. 3.2 provides theoretical explanations for the calculated differences within the averaging metrics. We then evaluate in Sect. 3.3 the discrepancies in the average aerosol pH raised by different temporal resolutions. At last, the conclusions of this study are given in Sect. 4.

## 2 Methods

### 2.1 Evaluation datasets

The datasets were obtained from atmospheric simulations with the three-dimensional GEOS-Chem chemical transport model (version 14.1.1, DOI: 10.5281/zenodo.1343546). The North China Plain (33°N−41°N, 114.375°E−120°E, Fig. S1) was chosen as the modeling region where multiple studies on aerosol pH have been conducted because of the concern about haze events. The vertical grid spanned from the surface to the mesosphere, encompassing 47 hybrid sigma/pressure levels. The horizontal resolution of 0.625° (longitude) × 0.5° (latitude) was used and the boundary conditions were supplied by a global simulation

with a coarser resolution of 5° × 4°. Meteorological input was from the Modern-Era Retrospective Analysis for Research and Applications, version 2 (MERRA-2) product, provided by the Goddard Earth Observing System (GEOS) of NASA's Global Modeling and Assimilation Office (Gelaro et al., 2017). The simulation period covered the winter season from December 2018 to February 2019 and the summer season from June to August 2019. The detailed settings of emission databases and chemical mechanisms are shown in Text S1. The modeled concentrations of fine aerosol components ($SO_4^{2-}$, $NO_3^-$, $NH_4^+$, elemental carbon, and organic materials) were evaluated with the Tracking Air Pollution in China (TAP) dataset (Geng et al., 2017; Liu et al., 2022; Wang et al., 2012; Wang et al., 2020a). As a reanalysis data product, TAP amalgamated surface observations, remote sensing, emission inventories, and model simulations to construct a near real-time dataset of aerosol and gas pollutant concentrations over China. We found a reasonable agreement between our GEOS-Chem model simulation results and the TAP reanalysis dataset (Fig. S2).

The ISORROPIA-II model (version 2.2) was used in GEOS-Chem to calculate the thermodynamic equilibrium processes for the $H^+$−$NH_4^+$−$K^+$−$Ca^{2+}$−$Mg^{2+}$−$Na^+$−$OH^-$−$SO_4^{2-}$−$NO_3^-$−$Cl^-$−$H_2O$ inorganic aerosol system (Fountoukis and Nenes, 2007; Pye et al., 2009). The model assumed that $\gamma_{H^+}$ was always equal to unity. The calculation of pH was simplified as Eq. (5)

$$\text{pH} = -\log_{10}\left(\frac{m_{H^+}}{m^\ominus}\right) \tag{5}$$

$$m_{H^+} = \frac{x_{H^+}}{x_{\text{water}}} \times 55.509 \tag{6}$$

where $x_{H^+}$ and $x_{\text{water}}$ indicated molar fraction of $H^+$ and aerosol liquid water, respectively. $m^\ominus$ was the standard molality (1 mol $kg^{-1}$ water), and 55.509 was the molality of water (Peng et al., 2019).

During the application of ISORROPIA-II, we assumed that the aerosol was internally mixed, forming a single aqueous phase encompassing the inorganic species, without phase separations that could affect pH (Guo et al., 2017). In the mode calculations, meteorological data (T and RH), gaseous concentrations (HCl, $HNO_3$, and $NH_3$), and aerosol concentrations ($SO_4^{2-}$, $NO_3^-$, $NH_4^+$, $Cl^-$, fine-sized dust, and fine-sized sea salt) were called by the ISORROPIA-II routines. Fine-sized dust was used to estimate $Ca^{2+}$ and $Mg^{2+}$, and fine-sized sea salt was used to estimate $Na^+$ and $Cl^-$ (Wang et al., 2019). We chose the forward mode (i.e., using the total gas + aerosol concentrations as model inputs) and assumed that the aerosol was in the metastable state. Calculations using only aerosol-phase composition as model inputs (i.e., reverse mode) have been suggested sensitive to observational errors of ionic species and thus should be avoided (Hennigan et al., 2015). The assumed particle phase state, either stable or metastable, does not significantly affect pH calculations (Song et al., 2018).

Two datasets from the GEOS-Chem simulation outputs were used in this study. The first dataset was used in subsections 3.1 and 3.2, which encompassed aerosol pH, AWC, meteorological data (T and RH), gaseous concentrations (HCl, $HNO_3$, and $NH_3$), and aerosol concentrations ($SO_4^{2-}$, $NO_3^-$, $NH_4^+$, fine-sized dust, fine-sized sea salt, and $PM_{2.5}$) at the surface layer during the 2018/2019 winter for the North China Plain. The AWC and aerosol pH values were calculated online within the GEOS-Chem model using the incorporated ISORROPIA-II thermodynamic module. The dataset had a temporal resolution of 3 hours,

consistent with that for the meteorological input data of GEOS-Chem. Following previous studies (Guo et al., 2016; Haskins et al., 2018), we selected the data with RH between 25% and 95% to meet the metastable assumption set by ISORROPIA-II and to avoid the large uncertainty associated with very high RH. After filtering, this dataset contained approximately 100,000 individual samples. The second dataset consisted of the chemical and meteorological data in the Beijing grid (centered at 40°N, 116.25°E) extracted from the first dataset. This dataset had the same temporal resolution of 3 hours and 720 samples for the

2018/2019 winter season. It could be considered as a pseudo-observation dataset analogous to what was reported by a field campaign.

**2.2 Statistical analysis**

We computed the following five data metrics describing the central tendency of a pH dataset: $pH_{Md}$ (median of pH), $pH_{Mo}$ (mode of pH), $\overline{pH}$ (arithmetic mean of pH), $\overline{pH}^*$ (pH based on the arithmetic mean of $a_{H^+}$), and $\overline{pH}_w^*$ (pH based on the

AWC-weighted mean of $a_{H^+}$). As per statistical definitions, $pH_{Md}$ and $pH_{Mo}$ represent the value of the 50th percentile and the most frequently occurring value of the dataset, respectively. The algorithms for calculating $\overline{pH}$, $\overline{pH}^*$, and $\overline{pH}_w^*$ have been provided in Eqs. (2−4), respectively. The probability density function of aerosol pH and AWC was calculated using the Kernel Smoothing Function (*ksdensity* and *mvksdensity*), a feature within the Statistics Toolbox of the MATLAB R2021b software. Prior to data processing, AWC was logarithmized. The "*ksdensity*" function was employed to calculate the probability density

functions of aerosol pH and AWC, respectively. Meanwhile, the "*mvksdensity*" function was employed to calculate the joint probability density function of the two variables. The data metrics for averaging pH were calculated utilizing Microsoft Excel 2016. In the calculations, a bandwidth of 0.1 was set to preserve important features of the distribution while suppressing noise.

To derive comprehensive distributional parameters from the available dataset and to construct appropriate confidence intervals to minimize statistical randomness, we used the Bootstrap approach (a statistical resampling technique, implemented

through the "*datasample*" function of the Statistics and Machine Learning Toolbox of the MATLAB R2021b software). In this study, our original dataset in winter comprised 105,403 sets of data (Section 3.1). We extracted 1,000 new datasets with 10,000 sets of data in each. We also conducted a similar sampling for the Beijing pseudo-observation data (720 sets of data, Section 3.3). Each dataset underwent 720 samplings and a total of 1,000 new datasets were collected. For each new dataset, calculate $\overline{pH}$, $pH_{Md}$, $pH_{Mo}$, $\overline{pH}^*$, and $\overline{pH}_w^*$ separately.

**3 Results and discussion**

**3.1 Distribution of aerosol pH and aerosol water content**

We present the probability distributions of the aerosol pH and AWC for the winter season in Fig. 1a and Fig. 1c, respectively,

as well as their joint probability distribution in Fig. 1b. It can be seen that the distributions of aerosol pH and AWC are not independent (Yuan and Shou, 2022). Mechanistic studies have revealed that AWC was a primary contributor to pH shifts. Zheng et al. (2020) proposed a multiphase buffer theory suggesting that AWC could considerably regulate the peak buffer pH of the individual buffering agent (i.e., conjugate acid-base pairs $NH_4^+/NH_3$, $HSO_4^-/SO_4^{2-}$, and $HNO_3/NO_3^-$). The distribution of AWC was characterized similarly to a skewed log-normal distribution, with noticeable differences between its arithmetic mean (53.3 μg m$^{-3}$), median (6.8 μg m$^{-3}$), and mode (0.5 μg m$^{-3}$). The properties of hygroscopic components in aerosols and the positive feedback between the primary hygroscopic components (SNA: sulfate, nitrate, and ammonium) and aerosol water content lead to an exponential response of AWC to changes in relative humidity (Liu et al., 2023; Zhang et al., 2021b; Wang et al., 2020b). Simultaneously, the ambient relative humidity typically exhibits a skewed normal distribution (Yuan et al., 2020). These factors collectively shape the probability distribution of AWC.

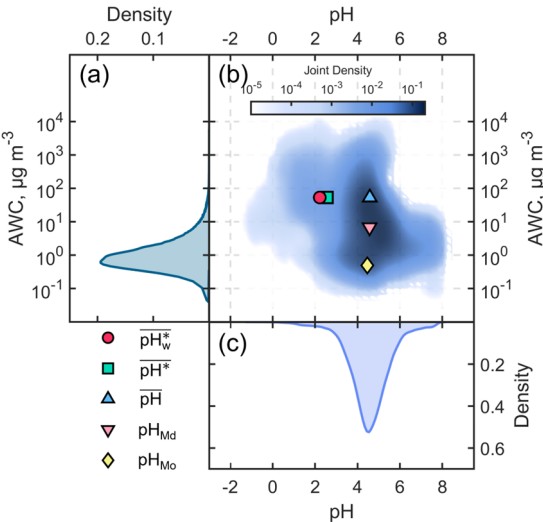

**Figure 1. Probability distributions of (a) aerosol water content (AWC, μg m$^{-3}$) and (c) aerosol pH, and (b) the joint probability distribution of AWC and aerosol pH in the North China Plain during winter season from December 2018 to February 2019. The position of the blue triangle is based on the $\overline{pH}$ and the $\overline{AWC}$, the pink inverted triangle is based on the $pH_{Md}$ and the $AWC_{Md}$, the yellow diamond is based on the $pH_{Mo}$ and the $AWC_{Mo}$, the green square is based on the $\overline{pH^*}$ and the $\overline{AWC}$, and red circle is based on $\overline{pH_w^*}$ and $\overline{AWC}$.**

On the other hand, the distribution of aerosol pH was approximate to a skewed normal distribution, along with very small differences (< 0.1 unit) among its arithmetic mean ($\overline{pH}$, 4.6), median ($pH_{Md}$, 4.6), and mode ($pH_{Mo}$, 4.5). However, the calculated $\overline{pH^*}$ (the arithmetic mean of pH based on $a_{H^+}$) was 2.6, close to 2 units lower than the above three metrics, indicating a deviation of around 2 orders of magnitude in the activity of hydrogen ions. $a_{H^+}$ followed a skewed log-normal distribution. Based on the AM-GM inequality (geometric mean does not exceed arithmetic mean), it can be deduced that $\overline{pH^*}$ is always less than or equal to $\overline{pH}$. For example, assuming that the aerosol was strong acidity for half of a day, such as pH=1, which means $a_{H^+}$ was $10^{-1}$; and the aerosol was weak acidity at the remaining time, such as pH=5, which meant $a_{H^+}$ was

$10^{-5}$. In this case, the $\overline{\text{pH}}$ was 3 ($\overline{\text{pH}} = \frac{1+5}{2} = -\log_{10}\sqrt{10^{-1}\cdot10^{-5}}$), while the $\overline{\text{pH}^*}$ was 1.3 ($\overline{\text{pH}^*} = -\log_{10}(\frac{10^{-1}+10^{-5}}{2})$), as is evident that $\sqrt{10^{-1}\cdot10^{-5}}$ is less than $\frac{10^{-1}+10^{-5}}{2}$. $\overline{\text{pH}_w^*}$ (AWC-weighted mean) differed from $\overline{\text{pH}^*}$ by only about 0.4 unit.

We employed the Bootstrap approach to measure the dispersion for $\overline{\text{pH}}$, $\text{pH}_{\text{Md}}$, $\text{pH}_{\text{Mo}}$, $\overline{\text{pH}^*}$, and $\overline{\text{pH}_w^*}$. We extracted 1,000 new datasets, each comprising 10,000 sets of data, and calculated $\overline{\text{pH}}$, $\text{pH}_{\text{Md}}$, $\text{pH}_{\text{Mo}}$, $\overline{\text{pH}^*}$, and $\overline{\text{pH}_w^*}$ for each new dataset separately. The results of the statistical analysis were shown in Fig. 2. The results indicated that the means of $\overline{\text{pH}}$, $\text{pH}_{\text{Md}}$, $\text{pH}_{\text{Mo}}$, $\overline{\text{pH}^*}$, and $\overline{\text{pH}_w^*}$ were 4.6, 4.6, 4.5, 2.6, and 2.2, respectively, which were consistent with our original dataset. Additionally, the results exhibited high stability, with minimal differences in interquartile distances, namely 0.13, 0.19, 0.02, 0.02, and 0.38, respectively.

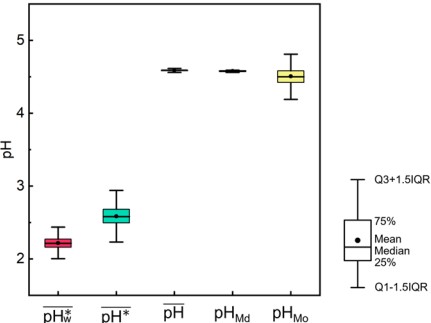

**Figure 2. Dispersion in the calculations of $\overline{\text{pH}}$, $\text{pH}_{\text{Md}}$, $\text{pH}_{\text{Mo}}$, $\overline{\text{pH}^*}$, and $\overline{\text{pH}_w^*}$ in the North China Plain winter 2018, based on Bootstrap. The results were extracted from 1,000 new datasets, each containing 10,000 sets of data. In the box–whisker plots, the points indicate means, the whiskers, and boxes indicate the values greater than the sum of the upper quartile and 1.5 times IQR, 75th percentiles, 50th percentiles, 25th percentiles, the values less than the sum of the lower quartile and 1.5 times IQR, respectively.**

Indeed, it's noteworthy that the aforementioned discrepancies were notably diminished during the summer season. Fig. S3 illustrates the probability distribution of aerosol pH and AWC during the summer season, as well as their joint probability distribution. The joint distribution in summer was opposite to the winter results, with higher pH observed at high AWC values and lower pH at low AWC values. This is because summer months are typically cleaner, with AWC predominantly influenced by RH. The resulting high AWC has a dilution effect on acidic components, leading to higher pH levels. The quantitative results for $\overline{\text{pH}}$, $\text{pH}_{\text{Md}}$, $\text{pH}_{\text{Mo}}$, $\overline{\text{pH}^*}$, and $\overline{\text{pH}_w^*}$ were 2.6, 2.7, 3.0, 2.0, and 2.4, respectively. The main reason for the lower pH in summer compared to winter is the temperature difference (Text S2). While $\overline{\text{pH}^*}$ and $\overline{\text{pH}_w^*}$ remained lower than $\overline{\text{pH}}$, the difference was significantly smaller compared to winter. The smaller range of pH also contributed to the proximity of the three statistics.

The significant bias between the averaging metrics calculated based on the pH value of individual samples ($\overline{\text{pH}}$, $\text{pH}_{\text{Md}}$, and $\text{pH}_{\text{Mo}}$) and those based on the activity of hydrogen ions of individual samples ($\overline{\text{pH}^*}$ and $\overline{\text{pH}_{\text{w}}^*}$) may have important implications on the understanding of atmospheric processes regulated by aerosol pH (Pye et al., 2020). For instance, the phase partition of $HNO_3-NO_3^-$ and $NH_3-NH_4^+$ could achieve a complete transition between the gaseous and particulate phases with two units of pH change in their sensitive regime (Chen et al., 2016; Chen et al., 2018).

**3.2 Variations of aerosol pH with relative humidity**

In order to further explain the discrepancies among different aerosol pH averaging metrics in winter season, we calculated the trend of $\overline{\text{pH}_{\text{w}}^*}$, $\overline{\text{pH}^*}$, $\overline{\text{pH}}$, $\text{pH}_{\text{Md}}$, and $\text{pH}_{\text{Mo}}$ with increasing RH bins (Fig. 3a). As shown, $\overline{\text{pH}}$, $\text{pH}_{\text{Md}}$, and $\text{pH}_{\text{Mo}}$ had a similar gradually decreasing trend with RH. $\overline{\text{pH}_{\text{w}}^*}$ and $\overline{\text{pH}^*}$, however, showed a different pattern from the above three metrics. Interestingly, there were significant drops in $\overline{\text{pH}_{\text{w}}^*}$ and $\overline{\text{pH}^*}$ when RH increased from 30% to 50%, which then remained nearly constant within the RH range from 50% to 90%. Fig. S4 showed the joint probability distribution of AWC and aerosol pH, mirroring Fig. 1b, but with RH intervals of 10%. The characteristic skewed log-normal distributions of RH in the range of 40% to 90% showed a right-skewed pH probability distribution (i.e., equal probability densities with a wider range for low pH). Conversely, RH = 30% aligned closer to a log-normal distribution where pH exhibited a symmetrical distribution. This explained the variation in the gaps of aerosol pH between statistical metrics across different RH levels in Fig. 3a.

We investigated the proportion changes of secondary inorganic aerosols (SNA: $NH_4^+$, $SO_4^{2-}$, and $NO_3^-$) and dust in $PM_{2.5}$ under different RH conditions. As depicted in Fig. 3b, in general, the elevation of RH was accompanied by an ascending in the fraction of SNA and a descent in that of dust. The large proportion of dust at the low RH (~30%) was believed to enhance aerosol pH (Guo et al., 2018). As the RH escalated to around 50%, the SNA proportion experienced a rapid ascent, concurrently with a precipitous decline in the dust proportion, which better compatibly explained the decrease in aerosol pH. The relatively stable $\overline{\text{pH}_{\text{w}}^*}$ variation at from 50% to 90% RH could be explained by the multiphase buffering theory (Zheng et al., 2020; Zheng et al., 2022). The theoretical equation derived from the multiphase buffering theory (see Text S2) suggested, when the aerosol pH was predominantly moderated by the buffering of the conjugate acid-base pair $NH_3/NH_4^+$, that aerosol pH could be simplified as a function of $p_{NH_3}$ (partial pressure of gaseous $NH_3$), $[NH_4^+(aq)]$ (molality of $NH_4^+$ in aerosol water), and $\gamma_{NH_4^+(aq)}$ (activity coefficient of $NH_4^+(aq)$). Fig. 3c illustrated that $p_{NH_3}$ was nearly constant under varying RH. Fig. 3d showed that $[NH_4^+(aq)]$ displayed a downward trend with elevated RH whereas $\gamma_{NH_4^+(aq)}$ exhibited an upward trend particularly during the 70%–90% RH range. Overall, in consideration of the alterations in $p_{NH_3}$, $[NH_4^+(aq)]$, and $\gamma_{NH_4^+(aq)}$, it

225 was understandable that $\overline{\mathrm{pH}_w^*}$ appeared to have minor variations within 50%−90% RH. Additionally, it should be noted that

the reason for the consistent trend of $\overline{\mathrm{pH}_w^*}$ and $\overline{\mathrm{pH}^*}$ with RH variation is that in most cases, there was more $H^+$ production

when AWC was high, and vice versa, as seen in both Fig. 1 and Fig. 3a. This led to the discrepancies between the $\overline{\mathrm{pH}_w^*}$ and

$\overline{\mathrm{pH}^*}$ caused by aerosol water content being masked.

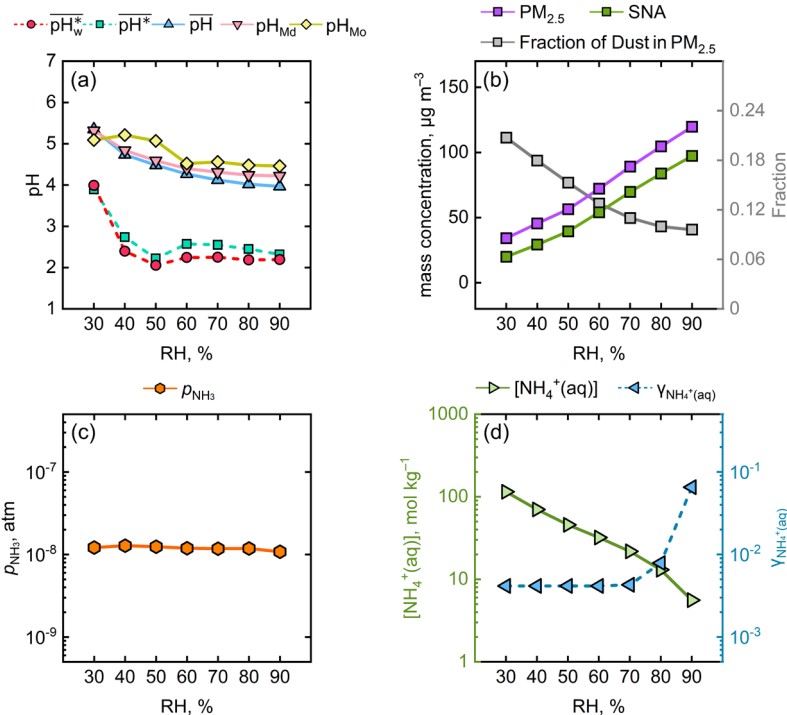

**Figure 3. Variations of several chemical and physical parameters as a function of RH. (a) $\overline{\mathrm{pH}_w^*}$, $\overline{\mathrm{pH}^*}$, $\overline{\mathrm{pH}}$, $\mathrm{pH_{Md}}$, and $\mathrm{pH_{Mo}}$. (b)**
**Fraction of SNA (the summation of sulfate, nitrate, and ammonium) and Dust in PM2.5. (c) $p_{\mathrm{NH_3}}$(atm). (d) $\left[\mathrm{NH_4^+(aq)}\right]$ (mol kg$^{-1}$)**
**and $\gamma_{\mathrm{NH_4^+(aq)}}$. The range of RH was 25%−95%, and the parameters were averaged at 10% RH intervals.**

235 **3.3 Influence of time resolution in input data on the averaged aerosol pH**

In our summary for aerosol pH calculations using the thermodynamic modeling approach and the data from chemical transport

model simulations and field observations (Table S1), we found that the temporal resolution of the input data varied substantially

ranging from less than an hour to a year. This inconsistency might arise from the differences in the native resolutions of field

sampling and model setting or from the choice of data aggregation in various studies. Differences in temporal resolution of the

240 data used in calculating pH may lead to disparate results, making it inappropriate to directly compare the average aerosol pH

values from different studies. In order to assess the impact of temporal resolutions, we applied the pseudo-observation data in

Beijing from the winter season from December 2018 to February 2019 as inputs to the ISORROPIA-II thermodynamic model.

The measures of dispersion for this site were shown in Fig. S5. The original input data with a 3-hour resolution, including both

chemical and meteorological variables, were aggregated into daily, weekly, monthly, and seasonal time steps. The results of $\overline{AWC}$, $\overline{pH}$, $\overline{pH^*}$, $\overline{pH_w^*}$, $pH_{Md}$, and $pH_{Mo}$ calculated based on corresponding ISORROPIA-II output are listed in Table 1.

**Table 1. Comparison of $\overline{AWC}$, $\overline{pH}$, $\overline{pH^*}$, $\overline{pH_w^*}$, $pH_{Md}$, and $pH_{Mo}$ calculated based on different temporal resolutions.**

| Temporal resolution | $\overline{AWC}$ (µg m$^{-3}$) | $\overline{pH}$ | $\overline{pH^*}$ | $\overline{pH_w^*}$ | $pH_{Md}$ | $pH_{Mo}$ |
|---|---|---|---|---|---|---|
| 3-hour | 10.0 | 5.1 | 3.2 | 2.1 | 4.8 | 4.4 |
| daily | 5.0 | 4.9 | 4.0 | 3.0 | 4.8 | 5.2 |
| weekly | 2.8 | 4.9 | 4.3 | 4.1 | 5.1 | 5.1 |
| monthly | 2.5 | 4.7 | 4.7 | 4.6 | 4.7 | — |
| seasonal | 2.5 | 4.6 | 4.6 | 4.6 | — | — |

The alteration by temporal resolutions exerted distinct degrees of influence on $\overline{AWC}$, $\overline{pH}$, $\overline{pH^*}$, $\overline{pH_w^*}$, $pH_{Md}$, and $pH_{Mo}$. $\overline{AWC}$ showed an overall declining trend as the time resolution became lower. This was mainly due to the fact that AWC had a general exponential relationship with RH and thus the high AWC incidences were largely averaged out when averaging RH at a lower resolution. The maximum deviation for $\overline{pH}$, $\overline{pH^*}$, $pH_{Md}$, and $pH_{Mo}$ were 0.5, 1.5, 1.4, and 0.3 units, respectively. Of particular note was that there was a maximum deviation of 2.5 units in $\overline{pH_w^*}$, suggesting approximately more than 2 orders of magnitude fluctuation in the activity of H$^+$. The discrepancy in $\overline{pH_w^*}$ was partly due to rapid and transient fluctuations in AWC during variations in meteorological conditions. $\overline{pH_w^*}$ tended to be lower as the temporal resolution got rougher which had an opposite trend with $\overline{pH}$. The results indicated that comparing the average aerosol pH metrics with non-uniform temporal resolutions might lead to erroneous conclusions.

Here, we use a simple example to illustrate the potential effect of temporal resolution in input data on multiphase chemistry reaction rates. It is well known that the rates of sulfate production from the oxidation of SO$_2$ by dissolved O$_3$ in the aqueous phase are pH dependen (Seinfeld and Pandis, 2016). The Beijing pseudo-observation data were applied with both 3-hour and daily resolutions to assess only the effect of deviations in aerosol pH on this sulfate formation pathway. The mean levels of SO$_2$ (3.8 ppb), O$_3$ (15.1 ppb), and T (269.8 K) of the study period (Dec 2018–Feb 2019) were used in the calculations. More detailed formulas are provided in Supplement Information (Tables S2 and S3). It was seen from Fig. S6 that calculating with daily-resolved data resulted in many rapid sulfate production incidences otherwise not being captured. We also discussed the sulfate formation rate d(SO$_4^{2-}$)/dt vs. AWC and d(SO$_4^{2-}$)/dt vs. pH to isolate the effect of AWC (Fig. S6 b-e). This example showed that aerosol pH had a greater effect on the rate of sulfate production than AWC, where aerosol pH had a linear

relationship with the $d(SO_4^{2-})/dt$. The mean sulfate formation rate for this winter period calculated based on 3-hour resolution data was 1.57 µg m$^{-3}$, while the corresponding value calculated based on the daily-resolved data was merely 0.72 µg m$^{-3}$, a factor of two smaller, indicating the significance of temporal resolution on the estimate of this chemical pathway.

## 4 Conclusions

In the present study, we evaluated the discrepancies in the average aerosol pH arose from differences in the averaging metrics and temporal resolutions based on thermodynamic modeling and evaluation datasets from a chemical transport model. Among the five metrics investigated ($\overline{pH}$, $pH_{Md}$, $pH_{Mo}$, $\overline{pH}^*$, and $\overline{pH}_w^*$), the former three metrics (calculated based on the pH value of individual samples) were found ~2 units higher than the latter two (based on the $a_{H^+}$ of individual samples) in winter, although there were only minor differences within each group. In summer, however, the differences were small for all five metrics. The change in the temporal resolutions of input data into thermodynamic models exerted distinct degrees of influence on the five metrics, with a maximum deviation of >2 units in $\overline{pH}_w^*$. The variation in $\overline{pH}_w^*$ was partly due to the fluctuations in aerosol water content.

Previous studies have highlighted the importance of maintaining consistency in terms of the assumed standard states (Jia et al., 2018) and the thermodynamic model used and the calculation method adopted (e.g., open vs. closed system and metastable vs. stable state) (Hennigan et al., 2015; Song et al., 2018) when comparing pH results across studies. This technical note underscores the importance of avoiding the default use of the "arithmetic mean" as the sole measure of "average". Additionally, it is also essential to consider the uncertainties introduced by the chosen averaging approach and temporal resolutions, which should be described clearly in future studies to ensure comparability of aerosol pH between models and/or observations. Using this study as an example, pH results for the 2018/2019 winter in the North China Plain were derived at 3-hour resolution through GEOS-Chem simulations. Measures of central tendency include: arithmetic mean ($\overline{pH}$, 4.6), median ($pH_{Md}$, 4.6), and mode ($pH_{Mo}$, 4.5), the arithmetic mean based on $a_{H^+}$ ($\overline{pH}^*$, 2.6), and the volume-weighted mean based on AWC and $a_{H^+}$ ($\overline{pH}_w^*$, 2.2). For further details, refer to *Code and data availability*.

From an atmospheric chemical perspective, $\overline{pH}_w^*$ may offer a more accurate representation of the average aerosol pH state. However, significant changes in pH can induce shifts in reaction rates, and utilizing any averaging method may fail to capture the reaction dynamics over extended time scales. Therefore, when utilizing pH datasets for theoretical calculations of reaction rates, we advocate for the utilization of hourly resolution data over longer-time resolution data.

**Appendix A: List of abbreviations.**

| Abbreviation | Definition | Unit |
|---|---|---|
| $a_{H^+}$ | the activity of H$^+$ in the standard state of a hypothetical ideal aqueous solution of standard molality and the reference state of an infinite dilute solution | dimensionless |
| AWC | aerosol liquid water content | $\mu g\ m^{-3}$ air |
| $m_{H^+}$ | the molality of H$^+$ | mol kg$^{-1}$ water |
| $m^{\ominus}$ | standard molality | 1 mol kg$^{-1}$ water |
| pH | the negative log (base 10) of H$^+$ activity | dimensionless |
| $pH_{Md}$ | the median of pH | dimensionless |
| $pH_{Mo}$ | the mode of pH | dimensionless |
| $\overline{pH}$ | the arithmetic mean of pH | dimensionless |
| $\overline{pH}^*$ | the negative log (base 10) of the arithmetic mean of H$^+$ activity | dimensionless |
| $\overline{pH}_w^*$ | the negative log (base 10) of the AWC-based weighted mean of H$^+$ activity | dimensionless |
| $\gamma_{H^+}$ | the activity coefficient of H$^+$ | dimensionless |

*Code and data availability.* The data is available at: https://github.com/shaojiesong/GC14.1.1_output_for_pH. The code used in this paper has been described in Section 2.2, please contact the corresponding author upon request if there are further needs.

*Supplement*. The supplement related to this article is available online at: https://doi.org/yyyy.

*Author contributions.* Shaojie Song initiated the study. Haoqi Wang and Xiao Tian carried out analysis and wrote the initial

draft. All authors helped interpret the data, provided feedback, and commented on the manuscript.

*Competing interests.* The authors declare that they have no conflict of interest.

*Financial support.* This work is supported by funding from the National Natural Science Foundation of China (42205109), the Natural Science Foundation of Tianjin City (22JCYBJC01330), and TianHe Qingsuo open research fund of TSYS in 2022 & NSCC-TJ.

*Acknowledgements.* We thank Li Zhang (Nankai University) and Xuan Wang (City University of Hong Kong) for helpful discussions.

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
