# Peer review of "Technical note: Influence of different averaging metrics and temporal resolutions on aerosol pH calculated by thermodynamic modeling"

_EGUsphere, 2024_

## Author Comment (AC1)

Note: Reviewer's comments are in *black italics*. Our replies are in blue. Changes in the manuscript are in red.

***RC1: 'Comment on egusphere-2024-479', Anonymous Referee #2, 18 Mar 2024***

*The pH of aerosols is a crucial parameter that significantly impacts the entire atmospheric chemical*
5 *process, and it is of utmost importance to accurately assess its value. This article primarily focuses on the issues of averaging methods and temporal resolution that are often overlooked in the process of calculating pH. It is found that these two factors can lead to differences in pH values ranging from 0.5 to 2 units, with the potential to affect the sulfate formation rate by up to two times. This is a meaningful study that provides important insights for our further understanding of the influencing*
10 *factors of pH. I agree with the publication of this article in ACP. There are a few minor issues that need to be addressed before that.*

Response: Thank you very much for the positive evaluation. We have carefully considered your suggestions and have responded to them by point-to-point below.

*(1) I did not see how the value of activity of H⁺ used in calculating pH was obtained. It would be*
15 *helpful to make this clearer in the article.*

Response: The ISORROPIA-II model (version 2.2) was used in GEOS-Chem to calculate the thermodynamic equilibrium processes for the $H^+-NH_4^+-K^+-Ca^{2+}-Mg^{2+}-Na^+-OH^--SO_4^{2-}-NO_3^--Cl^--H_2O$ inorganic aerosol system in this paper. Numerous simplifying assumptions in ISORROPIA-II were taken to increase
20 computational speed and numerical stability without substantially compromising rigor. These include $\gamma_{H^+}$ was assumed to be equal to unity because the activity coefficient routines were unable to calculate them explicitly (Fountoukis and Nenes, 2007). Then the calculation of pH can be simplified as:

$$pH = -\log_{10}(a_{H^+}) = -\log_{10}(\gamma_{H^+}\frac{m_{H^+}}{m^\ominus}) = -\log_{10}(\frac{m_{H^+}}{m^\ominus}) - \log_{10}(\gamma_{H^+}) = -\log_{10}(\frac{m_{H^+}}{m^\ominus}) \quad (1)$$

25
$$m_{H^+} = \frac{x_{H^+}}{x_{water}} \times 55.509 \quad (2)$$

where $a_{H^+}$ is the activity of H⁺ (dimensionless), $\gamma_{H^+}$ is the activity coefficient (dimensionless) of H⁺, $m_{H^+}$ is the molality (mol kg⁻¹ water) of H⁺, $m^\ominus$ is the standard molality (1 mol kg⁻¹ water), and $x_{H^+}$ and $x_{water}$ indicate molar fraction of H⁺ and aerosol liquid water. 55.509 is the molality of water (Peng et al., 2019).

30
The calculation of the activity of H⁺ was neglected in the original paper, and we have made the following additions to the revised manuscript (Section 2.1 Evaluation datasets, **Page 4, Lines 99−105):**

35 The ISORROPIA-II model (version 2.2) was used in GEOS-Chem to calculate the thermodynamic equilibrium processes for the $H^+-NH_4^+-K^+-Ca^{2+}-Mg^{2+}-Na^+-OH^--SO_4^{2-}-NO_3^--Cl^--H_2O$ inorganic aerosol system (Fountoukis and Nenes, 2007; Pye et al., 2009). The model assumed that $\gamma_{H^+}$ was always equal

to unity. The calculation of pH was simplified as Eq. (5)

$$pH = -\log_{10}\left(\frac{m_{H^+}}{m^\ominus}\right) \tag{5}$$

$$m_{H^+} = \frac{x_{H^+}}{x_{water}} \times 55.509 \tag{6}$$

where $x_{H^+}$ and $x_{water}$ indicated molar fraction of H$^+$ and aerosol liquid water, respectively. $m^\ominus$ was the standard molality (1 mol kg$^{-1}$ water), and 55.509 was the molality of water (Peng et al., 2019).

*(2) The authors have discussed how averaging methods and temporal resolution can lead to significant differences in pH. It would be more helpful if they could provide more specific recommendations on how to use these.*

The primary limitation of existing studies is the opaque methodology employed for pH averaging, often defaulting to the "arithmetic mean" as the sole measure of "average". Therefore, our primary recommendation is to provide detailed information about the averaging approach and temporal resolutions employed in specific studies. Additionally, we suggest that future studies include the original temporal resolution of pH data in supplementary information or publicly available datasets. We provide examples illustrating how to report pH based on the findings of this study.

While $\overline{pH}_w^*$ is more consistent with the central tendency of aerosol pH in the sense of atmospheric chemistry. Thus, calculations involving reaction kinetics necessitate short time steps, and any averaging approach introduces significant uncertainties. Therefore, for subsequent calculations, we recommend utilizing hourly-resolved data rather than longer time-resolved data.

We have made the following additions to the revised manuscript (Section 4 Conclusions, **Page 11, Lines 281−292**):

This technical note underscores the importance of avoiding the default use of the "arithmetic mean" as the sole measure of "average". Additionally, it is also essential to consider the uncertainties introduced by the chosen averaging approach and temporal resolutions, which should be described clearly in future studies to ensure comparability of aerosol pH between models and/or observations. Using this study as an example, pH results for the 2018/2019 winter in the North China Plain were derived at 3-hour resolution through GEOS-Chem simulations. Measures of central tendency included: arithmetic mean ($\overline{pH}$, 4.6), median (pH$_{Md}$, 4.6), mode (pH$_{Mo}$, 4.5), the arithmetic mean based on $a_{H^+}$ ($\overline{pH^*}$, 2.6), and the volume-weighted mean based on AWC and $a_{H^+}$ ($\overline{pH}_w^*$, 2.2). For further details, refer to *Code and data availability*.

From an atmospheric chemical perspective, $\overline{pH}_w^*$ may offer a more accurate representation of the central tendency of aerosol pH. However, significant changes in pH can induce shifts in reaction

75 rates, and utilizing any averaging method may fail to capture the reaction dynamics over extended time scales. Therefore, when utilizing pH datasets for theoretical calculations of reaction rates, we advocate for the utilization of hourly resolution data over longer-time resolution data.

*(3) It seems that the x-axis in Figure 2(d) differs from the other sub-figures.*

80 Thank you for your careful review. We have revised Figure 2 (now Figure 3 in the revised manuscript) accordingly (**Page 9**):

[Figure]

Figure 3. Variations of several chemical and physical parameters as a function of RH. (a) $\overline{\text{pH}_w^*}$,
85 $\overline{\text{pH}^*}$, $\overline{\text{pH}}$, $\text{pH}_{\text{Md}}$, and $\text{pH}_{\text{Mo}}$. (b) Fraction of SNA (the summation of sulfate, nitrate, and ammonium) and Dust in PM$_{2.5}$. (c) $p_{\text{NH}_3}$(atm). (d) $\left[\text{NH}_4^+(\text{aq})\right]$ (mol kg$^{-1}$) and $\gamma_{\text{NH}_4^+(\text{aq})}$. The range of RH was 25%−95%, and the parameters were averaged at 10% RH intervals.

90 **Reference**

Fountoukis, C. and Nenes, A.: ISORROPIA II: a computationally efficient thermodynamic equilibrium model for K$^+$–Ca$^{2+}$–Mg$^{2+}$–NH$^{4+}$–Na$^+$–SO$_4^{2-}$–NO$_3^-$–Cl$^-$–H$_2$O aerosols, Atmos. Chem. Phys., 7, 4639-4659, 10.5194/acp-7-4639-2007, 2007.

95 Peng, X., Vasilakos, P., Nenes, A., Shi, G., Qian, Y., Shi, X., Xiao, Z., Chen, K., Feng, Y., and Russell, A. G.: Detailed Analysis of Estimated pH, Activity Coefficients, and Ion Concentrations between the Three Aerosol Thermodynamic Models, Environ. Sci. Technol., 53, 8903-8913, 10.1021/acs.est.9b00181, 2019.

---

## Author Comment (AC2)

Note: Reviewer's comments are in *black italics*. Our replies are in blue. Changes in the manuscript are in red.

***RC2: 'Comment on egusphere-2024-479', Anonymous Referee #1, 27 Mar 2024***

*In this manuscript, Wang et al. quantity the inconsistencies in reported aerosol pH values that can arise from different averaging metrics and temporal resolutions of the pH calculation that are commonly used in field and modeling studies. They apply different averaging metrics and time resolutions to an ensemble of data generated from atmospheric chemistry model simulations for the North China Plain over the winter season. They show differences in the "average" pH of up to 2 units and emphasize the importance of specifying the averaging metrics and temporal resolution in the future. The paper highlights an important but oft-neglected point about comparing aerosol pH values reported by different studies. The paper is well-written and the conclusions are based on sound analysis.*

*The following are my concerns about the paper and suggestions for improvement:*

Response: Thank you sincerely for your favorable assessment. We have thoroughly reviewed your comments and suggestions and responded by point-to-point below.

*1. It would be better if the authors could recommend some best practices that future studies could follow when reporting average pH values, given that pH and $H^+$ molality are non-conserved quantities and their arithmetic mean has little physical meaning, but also considering the practicalities of field studies. Should field studies report other relevant data that would make it easier to compare their findings to other studies or models? The main recommendation of the manuscript is that studies should "clearly state their chosen averaging metrics," but it seems from Table S1 that most studies already do that. The submission would be more significant if the authors could be more insightful in their recommendations.*

Response: While we provide specific information about the chosen averaging metrics in Table S1, it is crucial to acknowledge a significant limitation of existing studies: the opacity of pH averaging methods, often defaulting to the "arithmetic mean" as the sole measure of "average/mean". This lack of clarity posed a challenge to data collection for this study. To mitigate the recurrence of such issues, we urge researchers in aerosol acidity to be mindful of the averaging methods employed and to furnish detailed information regarding these methods and temporal resolutions in their studies. It would have been beneficial if the authors included the original temporal resolution of the pH data in the provided supplementary information or made it available in publicly accessible datasets. As per your suggestion, we have incorporated this study as an example and included recommended practices that can be adopted in the manuscript.

The purpose of this paper is not to determine which representation of central tendencies is correct. As mentioned in this paper, arithmetic mean ($\overline{\mathrm{pH}}$) is the most commonly reported approach in existing studies, and the arithmetic mean based on $a_{\mathrm{H}^+}$ ($\overline{\mathrm{pH}^*}$) aligns more closely with the central trend of aerosol pH from an atmospheric chemistry perspective. However, calculations involving reaction kinetics necessitate short time steps, and any averaging method

introduces significant uncertainties. Therefore, our additional recommendation is to utilize hourly-resolved data instead of longer-resolved data in subsequent theoretical calculations.

We have made the following changes in the revised manuscript (Section 4 Conclusions, **Page 11, Lines 281−292**):

This technical note underscores the importance of avoiding the default use of the "arithmetic mean" as the sole measure of "average". Additionally, it is also essential to consider the uncertainties introduced by the chosen averaging approach and temporal resolutions, which should be described clearly in future studies to ensure comparability of aerosol pH between models and/or observations. Using this study as an example, pH results for the 2018/2019 winter in the North China Plain were derived at 3-hour resolution through GEOS-Chem simulations. Measures of central tendency included: arithmetic mean ($\overline{\mathrm{pH}}$, 4.6), median ($\mathrm{pH_{Md}}$, 4.6), mode ($\mathrm{pH_{Mo}}$, 4.5), the arithmetic mean based on $a_{\mathrm{H^+}}$ ($\overline{\mathrm{pH}^*}$, 2.6), and the volume-weighted mean based on AWC and $a_{\mathrm{H^+}}$ ($\overline{\mathrm{pH}^*_w}$, 2.2). For further details, refer to *Code and data availability*.

From an atmospheric chemical perspective, $\overline{\mathrm{pH}^*_w}$ may offer a more accurate representation of the central tendency of aerosol pH. However, significant changes in pH can induce shifts in reaction rates, and utilizing any averaging method may fail to capture the reaction dynamics over extended time scales. Therefore, when utilizing pH datasets for theoretical calculations of reaction rates, we advocate for the utilization of hourly resolution data over longer time resolution data.

*2. The submission would also be stronger if it included a discussion of the measures of dispersion, as they are important when statistically comparing different pH datasets.*

Response: We appreciate your suggestion and agree that discussing measures of dispersion is crucial for comparing different pH datasets. However, to clarify, the most conventional measures of dispersion for the winter 2018 aerosol pH dataset in the North China Plain are standard deviation or percentiles. It's worth noting that this study presents five statistics of central tendency ($\overline{\mathrm{pH}}$, $\mathrm{pH_{Md}}$, $\mathrm{pH_{Mo}}$, $\overline{\mathrm{pH}^*}$, and $\overline{\mathrm{pH}^*_w}$), and standard deviation and percentiles only account for the variability of $\overline{\mathrm{pH}}$.

To derive comprehensive distributional parameters from an existing dataset and construct appropriate confidence intervals to mitigate statistical randomness, we employ Bootstrap. Simply put, Bootstrap is a statistical resampling technique. It offers a robust and flexible approach for statistical inference, particularly when the underlying distribution of the data is unknown or complex, or when traditional parametric methods may not be applicable. In this study, our original dataset comprised 105,403 sets of data. We utilized the Statistics and Machine Learning Toolbox in Matlab to perform BootStrap resampling, extracting 1,000 new datasets with 10,000 sets of data in each. For each new dataset, calculate $\overline{\mathrm{pH}}$, $\mathrm{pH_{Md}}$, $\mathrm{pH_{Mo}}$, $\overline{\mathrm{pH}^*}$, and $\overline{\mathrm{pH}^*_w}$

separately. We conducted statistical analyses on the central tendency of these 1000 datasets (Fig. 2). The results indicated that the means of $\overline{pH}$, $pH_{Md}$, $pH_{Mo}$, $\overline{pH^*}$, and $\overline{pH_w^*}$ were 4.6, 4.6, 4.5, 2.6, and 2.2 respectively, which were consistent with our original dataset. Additionally, the results exhibited high stability, with small interquartile distances (75th quartile minus 25th quartile) of 0.01, 0.01, 0.16, 0.19, and 0.11 respectively.

[Figure]

Figure 2. Dispersion in the calculations of $\overline{pH}$, $pH_{Md}$, $pH_{Mo}$, $\overline{pH^*}$, and $\overline{pH_w^*}$ in the North China Plain winter 2018, based on Bootstrap. The results were extracted from 1,000 new datasets, each containing 10,000 sets of data. In the box–whisker plots, the points indicate means, the whiskers, and boxes indicate the values greater than the sum of the upper quartile and 1.5 times IQR, 75th percentiles, 50th percentiles, 25th percentiles, the values less than the sum of the lower quartile and 1.5 times IQR, respectively.

We also conducted similar sampling for the Beijing pseudo-observation data (3-hour resolution, totaling 720 sets of data) in subsection 3.3. Each dataset underwent 720 samplings and a total of 1,000 new datasets were collected. And the means of $\overline{pH}$, $pH_{Md}$, $pH_{Mo}$, $\overline{pH^*}$, and $\overline{pH_w^*}$ calculated for each of these 1,000 datasets were 5.1, 4.8, 4.3, 3.2, and 2.2, respectively (Fig. S5), slightly different from the Beijing pseudo-observation data (5.1, 4.8, 4.4, 3.2, and 2.1). The stability of the results was good, but higher than that of the North China Plain, with interquartile distances of 0.06, 0, 0.3, 0.3 and 0.4, respectively. The reason was that the sample size of this dataset was much smaller than that of the North China Plain, leading to greater variability in repeated random sampling.

[Figure]

Figure S5. Dispersion in the calculations of $\overline{pH}$, $pH_{Md}$, $pH_{Mo}$, $\overline{pH^*}$, and $\overline{pH_w^*}$ in the pseudo-observation data for Beijing in winter 2018, based on Bootstrap. The results were extracted from 1,000 new datasets, each containing 720 sets of data. In the box–whisker plots, the points indicate means, the whiskers, and boxes indicate the values greater than the sum of the upper quartile and 1.5 times IQR, 75th percentiles, 50th percentiles, 25th percentiles, the values less than the sum of the lower quartile and 1.5 times IQR, respectively.

In the revised manuscript, we have made the following changes in Section 2.2 Statistical analysis, **Page 5, Lines 138−144**:

To derive comprehensive distributional parameters from the available dataset and to construct appropriate confidence intervals to minimize statistical randomness, we used the Bootstrap approach (a statistical resampling technique, implemented through the "*datasample*" function of the Statistics and Machine Learning Toolbox of the MATLAB R2021b software). In this study, our original dataset in winter comprised 105,403 sets of data (Section 3.1). We extracted 1,000 new datasets with 10,000 sets of data in each. We also conducted a similar sampling for the Beijing pseudo-observation data (720 sets of data, Section 3.3). Each dataset underwent 720 samplings and a total of 1,000 new datasets were collected. For each new dataset, calculate $\overline{pH}$, $pH_{Md}$, $pH_{Mo}$, $\overline{pH^*}$, and $\overline{pH_w^*}$ separately.

In the revised manuscript, we have made the following changes Section 3.1 Distribution of aerosol pH and aerosol water content, **Page 7, Lines 175−182**, and also added **Figure 2**:

We employed the Bootstrap approach to measure the dispersion for $\overline{pH}$, $pH_{Md}$, $pH_{Mo}$, $\overline{pH^*}$, and $\overline{pH_w^*}$. We extracted 1,000 new datasets, each comprising 10,000 sets of data, and calculated $\overline{pH}$, $pH_{Md}$, $pH_{Mo}$, $\overline{pH^*}$, and $\overline{pH_w^*}$ for each new dataset separately. The results of the statistical analysis

were shown in Fig. 2. The results indicated that the means of $\overline{pH}$, $pH_{Md}$, $pH_{Mo}$, $\overline{pH^*}$, and $\overline{pH_w^*}$ were 4.6, 4.6, 4.5, 2.6, and 2.2, respectively, which were consistent with our original dataset. Additionally, the results exhibited high stability, with minimal differences in interquartile distances, namely 0.13, 0.19, 0.02, 0.02, and 0.38, respectively.

[Figure]

Figure 2. Dispersion in the calculations of $\overline{pH}$, $pH_{Md}$, $pH_{Mo}$, $\overline{pH^*}$, and $\overline{pH_w^*}$ in the North China Plain winter 2018, based on Bootstrap. The results were extracted from 1,000 new datasets, each containing 10,000 sets of data. In the box–whisker plots, the points indicate means, the whiskers, and boxes indicate the values greater than the sum of the upper quartile and 1.5 times IQR, 75th percentiles, 50th percentiles, 25th percentiles, the values less than the sum of the lower quartile and 1.5 times IQR, respectively.

In the revised manuscript, we have made the following changes in Section 3.3 Influence of time resolution in input data on the averaged aerosol pH, **Page 9, Line 241,** and also added **Figure S5**:

The measures of dispersion for this site were shown in Fig. S5.

[Figure]

145    Figure S5. Dispersion in the calculations of $\overline{pH}$, $pH_{Md}$, $pH_{Mo}$, $\overline{pH}^*$, and $\overline{pH}_w^*$ in the pseudo-observation data for Beijing in winter 2018, based on Bootstrap. The results were extracted from 1,000 new datasets, each containing 720 sets of data. In the box–whisker plots, the points indicate means, the whiskers, and boxes indicate the values greater than the sum of the upper quartile and 1.5 times IQR, 75th percentiles, 50th percentiles, 25th percentiles, the values less

150    than the sum of the lower quartile and 1.5 times IQR, respectively. The means of $\overline{pH}$, $pH_{Md}$, $pH_{Mo}$, $\overline{pH}^*$, and $\overline{pH}_w^*$ calculated for each of these 1,000 datasets were 5.1, 4.8, 4.3, 3.2, and 2.2, respectively, slightly different from the Beijing pseudo-observation data (5.1, 4.8, 4.4, 3.2, and 2.1). The stability of the results was good, but higher than that of the North China Plain, with interquartile distances of 0.06, 0, 0.3, 0.3 and 0.4, respectively.

155

*3. The study used model data for the winter season. How different would the variations in "average" pH values be in the summer?*

Response: Thanks for your suggestion. We have incorporated the summer simulation results and computed the five "averages". In complete contrast to the winter results, the joint distribution in
160    summer was opposite to the winter results, with higher pH observed at high AWC values and lower pH at low AWC values (Fig. S3). This is because summer months are typically cleaner, with AWC predominantly influenced by RH. The resulting high AWC has a dilution effect on acidic components, leading to higher pH levels. The quantitative results for $\overline{pH}$, $pH_{Md}$, $pH_{Mo}$, $\overline{pH}^*$, and $\overline{pH}_w^*$ were 2.6, 2.7, 3.0, 2.0, and 2.4, respectively. The main reason for the lower pH in
165    summer compared to winter is the difference in temperature (Text S3). While $\overline{pH}^*$ and $\overline{pH}_w^*$ remained lower than $\overline{pH}$, the difference was significantly smaller compared to winter. The smaller range of pH also contributed to the proximity of the three statistics.

We have made several modifications in the main text to underscore that the methods employed
170    in this paper are generic. However, it is essential to recognize that results may vary depending on the location and time period of the study.

**Page 4, Line 91:**

The simulation period covered the winter season from December 2018 to February 2019 and the summer season from June to August 2019.

**Page 7, Lines 188−196:**

Indeed, it's noteworthy that the aforementioned discrepancies were notably diminished during the summer season. Fig. S3 illustrates the probability distribution of aerosol pH and AWC during the summer season, as well as their joint probability distribution. The joint distribution in summer was opposite to the winter results, with higher pH observed at high AWC values and lower pH at low AWC values. This is because summer months are typically cleaner, with AWC predominantly influenced by RH. The resulting high AWC has a dilution effect on acidic components, leading to higher pH levels. The quantitative results for $\overline{pH}$, $pH_{Md}$, $pH_{Mo}$, $\overline{pH^*}$, and $\overline{pH^*_w}$ were 2.6, 2.7, 3.0, 2.0, and 2.4, respectively. The main reason for the lower pH in summer compared to winter is the temperature difference (Text S3). While $\overline{pH^*}$ and $\overline{pH^*_w}$ remained lower than $\overline{pH}$, the difference was significantly smaller compared to winter. The smaller range of pH also contributed to the proximity of the three statistics.

**Page 11, Lines 271−276:**

In the present study, we evaluated the discrepancies in the average aerosol pH arose from differences in the averaging metrics and temporal resolutions based on thermodynamic modeling and evaluation datasets from a chemical transport model. Among the five metrics investigated $\overline{pH}$, $pH_{Md}$, $pH_{Mo}$, $\overline{pH^*}$, and $\overline{pH^*_w}$, the former three metrics (calculated based on the pH value of individual samples) were found ~2 units higher than the latter two (based on the $a_{H^+}$ of individual samples) in winter, although there were only minor differences within each group. In summer, however, the differences were small for all five metrics.

[Figure]

Figure S3. Same as Fig. 1, but for summer (June, July, and August) 2019. Probability distributions of (a) aerosol water content (AWC, µg m$^{-3}$) and (c) aerosol pH, and (b) the joint probability distribution of AWC and aerosol pH. The position of the blue triangle is based on the $\overline{pH}$ and the $\overline{AWC}$, the pink inverted triangle is based on the $pH_{Md}$ and the $AWC_{Md}$, the yellow diamond is based on the $pH_{Mo}$ and the $AWC_{Mo}$, the green square is based on the $\overline{pH}^*$ and the $\overline{AWC}$, and red circle is based on $\overline{pH}_w^*$ and $\overline{AWC}$.

*4. Fig. 1 shows several pH values above 6. Does ISORROPIA consider carbonate equilibrium in this case? If not, it is better not to include these values in the analysis.*

Response: Thank you for your comments. Several studies have demonstrated that carbonates exhibit a buffering effect on atmospheric acidity, particularly in cloud water (Shah et al., 2020; Pye et al., 2020). However, Zheng et al. (2023) has recently shown, based on multiphase buffering theory, that the buffering effect of carbonates on aerosol acidity is generally negligible compared to the multiphase buffering effect of ammonia. Their study demonstrated that within the range of aerosol liquid water content variation (from $10^{-6}$ to $5\times10^{-4}$ g m$^{-3}$), the corresponding p$Ka^*$ of $CO_2/HCO_3^-$ at 298 K was 15.8–18.4 (Fig. R1a, gray line), and this value was minimally affected by temperature. Based on the method, the p$Ka^*$ consistent with the present study (aerosol water content from $10^{-7}$ to $10^{-2}$ g m$^{-3}$) was calculated to be 14.4–19.4 at 298 K. Essentially, the buffering effect of carbonates becomes significant only when the aerosol pH exceeds 14 in the North China Plain, which is much higher than typical aerosol pH ranges.

[Figure]

Figure R1. Importance of inorganic carbon systems in buffering the atmospheric water. Here, we assume a constant $CO_2$ of 410 ppm. (a) Variation of the pKa* of $H_2CO_3/HCO_3^-$, $HCO_3^-/CO_3^{2-}$ in comparison with that of $NH_4^+/NH_3$ with liquid water content Lw at 298 K. This figure was taken from Zheng et al. (2023).

**Reference**

Pye, H. O. T., Nenes, A., Alexander, B., Ault, A. P., Barth, M. C., Clegg, S. L., Collett Jr, J. L., Fahey, K. M., Hennigan, C. J., Herrmann, H., Kanakidou, M., Kelly, J. T., Ku, I. T., McNeill, V. F., Riemer, N., Schaefer, T., Shi, G., Tilgner, A., Walker, J. T., Wang, T., Weber, R., Xing, J., Zaveri, R. A., and Zuend, A.: The acidity of atmospheric particles and clouds, Atmos. Chem. Phys., 20, 4809-4888, 10.5194/acp-20-4809-2020, 2020.

Shah, V., Jacob, D. J., Moch, J. M., Wang, X., and Zhai, S.: Global modeling of cloud water acidity, precipitation acidity, and acid inputs to ecosystems, Atmos. Chem. Phys., 20, 12223-12245, 10.5194/acp-20-12223-2020, 2020.

Zheng, G., Su, H., and Cheng, Y.: Role of Carbon Dioxide, Ammonia, and Organic Acids in Buffering Atmospheric Acidity: The Distinct Contribution in Clouds and Aerosols, Environ. Sci. Technol., 57, 12571-12582, 10.1021/acs.est.2c09851, 2023.